# Soil Type Rather Than Freezing Determines the Size of Soil-Root Plate of Silver Birch (*Betula pendula* Roth.) in the Eastern Baltic Region

**Oskars Krišāns [1], Roberts Matisons [1], Jānis Vuguls [1], Andris Seipulis [1], Didzis Elferts [1,2], Valters Samariks [1], Renāte Saleniece [1] and Āris Jansons [1,\*]**

1   Latvian State Forest Research Institute 'Silava', 111 Rigas Street, LV-2169 Salaspils, Latvia; oskars.krisans@silava.lv (O.K.); roberts.matisons@silava.lv (R.M.); janis.vuguls@silava.lv (J.V.); andris.seipulis@silava.lv (A.S.); didzis.elferts@lu.lv (D.E.); valters.samariks@silava.lv (V.S.); renate.saleniece@silava.lv (R.S.)
2   Faculty of Biology, University of Latvia, 1 Jelgavas Street, LV-1004 Riga, Latvia
\*   Correspondence: aris.jansons@silava.lv; Tel.: +371-2-910-9529

**Abstract:** In the Eastern Baltic region, severe windstorms increase both in frequency and magnitude, particularly during the dormancy period, increasing wind damage risks even more for silver birch (*Betula pendula* Roth.), which is considered to be less vulnerable forest tree species. Tree anchorage, particularly the properties of soil–root plate, determines the type of fatal failures trees experience under extreme wind loads and, subsequently, the potential for timber recovery during salvage logging. The link between soil–root plate properties and fatal failure types was assessed by conducting destructive static pulling tests; trees on freely draining minerals and drained deep peat soils under frozen and non-frozen soil conditions were tested. The size of the root plate did not differ between trees experiencing uprooting or stem breakage but was largely affected by soil type. Frozen soil conditions increased soil–root anchorage (via binding between soil particles) and, hence, the frequency of stem breakage without changing the size of soil–root plate. However, the lack of frozen soil conditions is among the main climatic risks for forestry within the region. The differences in the properties of soil–root plate implies plasticity in adaptation to wind loadings relative to birch, suggesting a potential for managing different types of fatal failure of trees and, subsequently, the share of retrievable timber in cases of salvage logging.

**Keywords:** soil–root plate; wind resistance; wind damage; frozen soil; deciduous forest; forest adaptation





## 1. Introduction

In the Eastern Baltic region, the ongoing warming and rising precipitations increase the vulnerability of forests relative to critical wind damage during late autumn and early spring [1–5], affecting also relatively wind-tolerant tree species, e.g., silver birch (*Betula pendula* Roth.) [3,6,7]. Tree wind loading resistance largely depends on soil–root anchorage, which is determined by the volume of the soil–root plate and by mutual binding between soil and roots [8,9], which, furthermore, can be enhanced by frozen soil conditions [9]. Under enhanced soil–root anchorage, tree mechanical stability becomes stem-strength dependent [10], leading to an increased frequency of stem breakage during critical mechanical loading and, therefore, amplifying economic losses [11]. However, trees with better anchorage are recognized to be more load-tolerant [12,13]; hence, the soil–root plate volume can be considered as a proxy of the vulnerability of trees to wind, which can aid the planning of adaptive forest management [14]. Such information can be acquired by static tree-pulling tests [9,12,13].

Although it is a well-known fact that trees tend to break more frequently under frozen soil conditions [8,9], to the authors' knowledge, the soil–root plate size for trees suffering

stem breakage has not been investigated. Therefore, this study aimed to compare soil–root plates of middle-aged silver birch according to the type of fatal failure either by uprooting or stem breakage on freely draining mineral soils and drained deep-peat soils under frozen and non-frozen conditions. To accomplish the aim, the artificial uprooting of trees that experienced stem breakage was performed. We hypothesized that trees experiencing stem breakage have larger soil–root plates. We also assumed that increased binding between soil and roots under the freezing conditions increased the soil–root plate.

## 2. Materials and Methods

### 2.1. Study Sites, Sample Trees, and Measurements

During January–March of 2021, under frozen and non-frozen soil conditions, 42 silver birches from 7 middle-aged (33 and 51 years) naturally regenerated stands on freely draining mineral and drained deep peat soils in the central part of Latvia (56°40′ N; 25°55′ E) on lowland terrain were tested. The study area is located in a hemiboreal forest zone with humid continental climate [15]. The mean monthly temperature ranges from $-4.5$ °C to 17.6 °C in January and July, respectively [16]. The soil freezing depth ranges 20–50 cm for nearly 40 days, although winters without freezing soil are becoming more frequent due to climate warming [16]. The mean annual sum of precipitation is 698.1 mm [16]. The wind climate is determined by westerlies [17] with the mean annual wind speed of 2.5 m·s$^{-1}$ [16] and the mean maximum wind speed at an elevation of 10 m of 17.2 m·s$^{-1}$ [18].

The studied birch stands had an admixture of Norway spruce (*Picea abies* (L.) H. Karst.), mostly in understory areas (14–40% of basal area). The mean diameter at breast height and the height of canopy trees were $19.6 \pm 0.6$ cm and $22.7 \pm 0.6$ m on mineral and $20.4 \pm 0.2$ cm and $22.6 \pm 0.7$ m on peat soil, respectively. The basal area of stands ranged 14.4–27.9 m$^2$·ha$^{-1}$ with stands on peat soils being generally denser. For testing, visually healthy and undamaged canopy trees representing the stem diameter distribution of stands were selected. Trees growing on the edges of openings, as well as close to each other, were avoided. Each stand was sampled under frozen and non-frozen soil conditions.

Prior the pulling, trees were de-topped 1 m over the half of height due to work safety considerations (Supplementary Materials, Figure S1). Trees were pulled until failure—either stem breakage or uprooting—using a pulley block system (Roll Double pulleys, Edelrid, Germany) attached on the half height of sample tree. Pulling force was applied by a motor winch (1800 Capstan Cable Winch, Nordforest, Germany) (Supplementary Materials, Figure S2). Trees that broke were re-anchored at the highest possible point and pulled once again until an overturn of the soil–root plate (uprooting) occurred. For each soil–root plate, the maximum depth was measured by piercing a steel rod near stem base, and five radii (at 0°, 45°, 90°, 135°, and 180°) of the surface of the soil–root plate were assessed (Supplementary Materials, Figure S3). The depth of frozen soil layer was measured at the edge of the uprooted soil–root plate at five points representing the radii.

### 2.2. Data Analysis

The volume of the soil–root plate (V$_{ROOTS}$) was estimated according to Krišāns et al. (2022) [10] as the volume of an elliptical paraboloid as follows:

$$V_{ROOTS} = \left(\frac{1}{2}\right) \cdot \pi \cdot a \cdot b \cdot h, \tag{1}$$

where h is the depth, and a and b are the longest and shortest of the five measured radii of the root plate, respectively.

Stemwood volume (V$_{STEM}$) was calculated as a proxy for the tree size using a locally developed equation [10]:

$$V_{STEM} = 0.0000909 \ldots H^{0.71677} \ldots DBH^{0.16692 \cdot 0.4343 \cdot \ln(H) + 1.7570} \tag{2}$$

which includes tree height (H) and stem diameter at breast height (DBH).

The differences in soil–root plate dimensions between trees experiencing stem breakage or uprooting, under frozen/non-frozen soil conditions, and the relevant soil type were estimated using a linear mixed-effects model:

$$y_{ij} = \mu + f_{ij} + sf_{ij} + s_{ij} + (site_j) + \varepsilon, \tag{3}$$

where $y_{ij}$ is $V_{ROOTS}$ expressed per $V_{STEM}$ to integrate tree size, $f_{ij}$ is a fixed effect of initial failure type (uprooting or stem breakage), $sf_{ij}$ is a fixed effect of frozen/non-frozen soil conditions, and $s_{ij}$ is a fixed effect of soil type (freely draining mineral or drained deep peat). Site was included as a random effect ($site_j$) to account for the uneven sample size. Due to limited sample size, interactions were not tested. The significance of fixed effects was estimated by Wald's $\chi^2$ test (considering that fixed effects had only two levels, the results also apply to their comparison). The statistical analysis of data was performed in R software (version 4.1.0.) [19], using packages "MuMIn" (model evaluation) [20], "emmeans" (comparison of levels of significant effects) [21], and "lme4" (model fit) [22].

## 3. Results and Discussion

In the Eastern Baltic region, middle-aged silver birch had significantly ($p < 0.001$) larger soil–root plate on drained deep peat soil than on freely draining mineral soil (Figure 1; Tables 1 and 2) due to larger surface radii (width). This indicated stronger leverage and, thus, adaptation to the damping of wind loads on considerably less stable soils [23–25]. Moreover, under non-frozen soil conditions, birch experienced more frequent stem breakage (exceeding uprooting) on peat soil, suggesting stronger soil–root anchorage due to larger and wider soil–root plates.

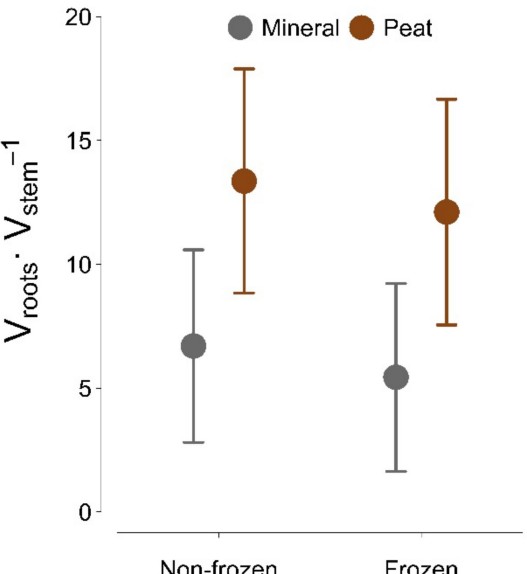

**Figure 1.** The estimated marginal mean (±95% confidence interval) volume of soil–root plate per stemwood volume for silver birch on freely draining mineral and drained deep peat soils under frozen and non-frozen conditions.

However, on an individual tree level within a stand, the size of the soil–root plate did not determine the type of failure (uprooting or stem breakage), as indicated by a non-significant ($p = 0.68$) relationship between failure type and the size of the soil–root plate. Although better anchored trees are considered to have higher loading resistances [12,13], such results support the notion that there has been no clear evidence that stem breakage requires higher loading than uprooting [12–14,26,27]. Unfortunately, this implies that the weak spot affecting failure types still cannot be predicted within a stand, resulting in an uncertainty in collective stability [10,28,29].

**Table 1.** Mean (±standard error) height (H), stem diameter at breast height (DBH), volume of stemwood ($V_{STEM}$) and soil–root plate ($V_{ROOTS}$), depth and width of soil–root plate, the total number (Tree *n*) and number of sampled silver birch trees with stem breakage, and frozen soil depth of freely draining mineral and drained deep-peat soils.

| Soil | H (m) | DBH (cm) | $V_{STEM}$ (m$^3$) | $V_{ROOTS}$ (m$^3$) | Depth (m) | Width (m) | Tree *n* | Stem Breakage *n* | Frozen Soil Depth (cm) |
|---|---|---|---|---|---|---|---|---|---|
| **Mineral** | | | | | | | | | |
| Frozen | 21.9 ± 1.2 | 18.8 ± 1.5 | 0.31 ± 0.07 | 1.43 ± 0.33 | 0.67 ± 0.16 | 1.72 ± 0.04 | 9 | 7 | 34 ± 1 |
| Non-frozen | 22.5 ± 0.5 | 19.8 ± 0.7 | 0.33 ± 0.03 | 2.17 ± 0.31 | 0.71 ± 0.11 | 2.12 ± 0.04 | 18 | 0 | — |
| **Peat** | | | | | | | | | |
| Frozen | 22.3 ± 1.5 | 22.3 ± 2.9 | 0.45 ± 0.14 | 5.16 ± 1.36 | 0.70 ± 0.40 | 3.09 ± 0.06 | 5 | 3 | 20 ± 2 |
| Non-frozen | 22.6 ± 0.6 | 20.4 ± 1.3 | 0.35 ± 0.05 | 4.50 ± 0.68 | 0.74 ± 0.15 | 2.97 ± 0.03 | 10 | 6 | — |

**Table 2.** Model estimates (Est., log scale), strength (Wald's $\chi^2$) and significance (*p*-value) of frozen and non-frozen soil, type of failure and soil on soil–root plate volume per tree size (stemwood volume) ($V_{ROOTS} \cdot V_{STEM}^{-1}$), and random effect of site and model performance (R$^2$) for silver birch on freely draining mineral and drained deep peat soils.

| | $V_{ROOTS} \cdot V_{STEM}^{-1}$ | | |
|---|---|---|---|
| **Predictors** | **Est.** | **$\chi^2$** | **$p$-Value** |
| (Intercept) | 1.83 | 69.00 | **<0.001** |
| Failure type | −0.08 | 0.17 | 0.68 |
| Frozen soil | −0.22 | 1.59 | 0.21 |
| Soil type | 0.74 | 11.63 | **<0.001** |
| **Random Effects** | | | |
| $\sigma^2$ | 0.19 | | |
| $\tau_{00}$ | 0.04 | | |
| ICC | 0.17 | | |
| $N_{site}$ | 7 | | |
| Observations | 42 | | |
| Marginal R$^2$ | 0.39 | | |
| Conditional R$^2$ | 0.49 | | |

Freezing conditions are known to enhance soil–root anchorage [9]; accordingly, the frequency of stem breakage increased, particularly on mineral soil, and became the dominant failure type while soil was frozen (Table 1). Although one might deduce that the freeze-binding of soil particles [8] would enhance the size (width) of the soil–root plate and, hence, anchorage, thus limiting uprooting. However, the soil–root plate of the trees that experienced stem breakage, as assessed after overturning the initially broken trees, did not differ significantly (*p* = 0.21) from the uprooted trees (Figure 1; Table 2). Instead, a marginal decrease in the root plate size was observed for both soil types, which might be related to higher soil stiffness and more likely causes roots to break closer to the center of the soil–root plate (at larger root diameter), as they have to crush open frozen soil [30]. Such effects were apparently weaker on peat soil due to wider a soil–root plate and shallower soil freezing depth (Table 2), implying lower uncertainties, while supporting the suitability of birch on unstable soils under increasing wind effects [31].

In the Eastern Baltic region, enhanced soil–root plates on drained deep peat soil increased the stability of trees regardless of soil freezing conditions, suggesting a high adaptability of silver birch to wind loading on unstable soil, such as drained deep peat [31]. However, this can be attributed at the stand level only, as no differences in soil–root plate size were observed for individual trees (within a stand). Furthermore, microsite conditions are known to greatly affect individual tree stability [12,32,33], particularly in naturally regenerated stands [34,35], thus maintaining uncertainties.

## 4. Conclusions

The hypothesis of the study was rejected, as the size of soil–root plates of Eastern Baltic silver birch experiencing stem breakage and uprooting were similar, irrespective of the frozen soil conditions. Substantially wider soil–root plates on drained deep peat soil suggests plastic stability adaptations of silver birch relative to mechanically unstable soil conditions. However, the size of the soil–root plate cannot be applied as a stability proxy at an individual tree level, but it could be successfully attributed to a stand level when evaluating tree suitability relative to certain growing conditions. Nevertheless, the dependency of failure type of silver birch to soil type suggested a potential for managing possible consequences of wind damages in terms of timber recovery during salvage logging, which is higher for uprooted trees.

**Supplementary Materials:** The following supporting information can be downloaded at: https://www.mdpi.com/article/10.3390/su14127332/s1, Figure S1: A scheme of the setup of static tree pulling, Figure S2: The location of motor winch during static tree pulling, Figure S3: Measurements of the radii of the surface of soil–root plate in 5 directions at $0°$, $45°$, $90°$, $135°$, and $180°$.

**Author Contributions:** Conceptualization, Ā.J., O.K. and J.V.; methodology, J.V., D.E. and R.M.; formal analysis, V.S. and D.E.; data curation, A.S., J.V. and R.S.; writing—original draft preparation, V.S. and J.V.; writing—review and editing, R.M., O.K. and Ā.J.; supervision, Ā.J.; project administration, Ā.J.; funding acquisition, Ā.J. All authors have read and agreed to the published version of the manuscript.

**Funding:** This research was funded by European Regional Development Fund project "Birch and aspen stand management decision support tool for reduction of wind damages" (No 1.1.1.1/18/A/134).

**Institutional Review Board Statement:** Not applicable.

**Informed Consent Statement:** Not applicable.

**Data Availability Statement:** Not applicable.

**Acknowledgments:** Authors acknowledge the valuable comments inspiring the work by Jānis Donis and help in field studies provided by Ilvars Ieviņš and Rolands Kāpostiņš.

**Conflicts of Interest:** The authors declare no conflict of interest.

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
