# Peer review of "Soil Type Rather Than Freezing Determines the Size of Soil-Root Plate of Silver Birch (Betula pendula Roth.) in the Eastern Baltic Region"

_sustainability, doi:10.3390/su14127332_

Round 1
Reviewer 1 Report
1) There are significant issues in relation to the statitical analysis:
- The statistical model used by the authors does not look coherent to me. They propose:
Root volume = f (type of failure, soil conditions, soil type)
but my suggestion is that the model that should be analyzed is:
Type of failure = f(root volume, soil conditions, soil type)
It is is root volume that influences the type of failure and not the other way round. Therefore, “type of failure” should be the dependent variable and “root volume” should be an independent variable.
Once the model has been changed, the authors should review section 3 in relation to the new results obtained
- The authors should clearly state the purpose of using each of the three R packages described in line 98.
- In lines 104-106 the authors compare the size of the root plate in peat and in mineral soils, but they do not provide any information in section 2.2 about the method used to compare them and to establish the significance of this comparison
- Similarly, it is not at all clear how the comparions in lines 128-120 of root-plate sizes between soil types was performed. The authors refer to Table 1, but this table does not compare those two variables.
2) The manuscript requires a significant revision in terms of written English
3) The dependent variable in Equation (1) (line 83) should be “VROOTS” and not just “V”
4) References should be completed, particularly numbers 4, 14, 16, 20, 24, and 33
Author Response
Reviewer 1
Point 1
The statistical model used by the authors does not look coherent to me. They propose:
Root volume = f (type of failure, soil conditions, soil type) but my suggestion is that the model that should be analysed is: Type of failure = f(root volume, soil conditions, soil type). It is root volume that influences the type of failure and not the other way around. Therefore, “type of failure” should be the dependent variable and “root volume” should be an independent variable. Once the model has been changed, the authors should review section 3 in relation to the new results obtained
Response 1
Indeed, from the first look on the introduction, one might get a feeling that factors affecting type of tree failure might be in the focus. However, this was not the case in our study, as it is a well-known fact that frozen soil conditions cause stem breakage more frequently than uprooting. We also converted the model as suggested by the reviewer and got trivial and expected result, as failure type was predicted by frozen soil conditions. Accordingly, we focus on comparing soil-root plate size between trees experiencing different failure types. In this study, we assessed size of soil-root plate for trees forced to uproot after stem breakage under frozen soil conditions.
Lines 48–49, we now state: “To accomplish the aim, artificial uprooting of trees that experienced stem breakage was performed.”
We however agree that pervious statement in the introduction did not lead the reader to necessity to form of such a comparison, hence it is now revised.
Lines 43–45, we now state: “Although it is a well-known fact that trees break rather than uproot under frozen soil conditions [8,9], to the authors’ knowledge, the differences in soil-root plate size for trees suffering uprooting or stem breakage have not been investigated.”
Point 2
The authors should clearly state the purpose of using each of the three R packages described in line 98.
Response 2
Such statement is difficult as the packages are mutually supportive and all of them are necessary for development and description of the statistical model used in this study.
Lines 102–104, we now clarify: “Data statistical analysis was done in R software (version 4.1.0.) [19], fitting and description of the statistical model was done using the packages “MuMIn” [20], “emmeans” [21], and “lme4” [22].”
Point 3
In lines 104-106 the authors compare the size of the root plate in peat and in mineral soils, but they do not provide any information in section 2.2 about the method used to compare them and to establish the significance of this comparison.
Response 3
Indeed, description of Post-Hoc test used to compare levels of significant factors would be of importance if any of them would have more than two levels or an interaction would have been assessed, which was not the case of this study. In case, significant factors have only two levels, the results of a Post-Hoc test would be identical to those presented in model ANOVA. Accordingly, the Post-Hoc test was not presented as it was unnecessary. We, however, agree that levels were not presented for all factors, hence they are now given.
Lines 99–100, we now clarify: ”…and sij is a fixed effect of soil type (freely draining mineral or drained deep peat).”
Lines 100–102, we now clarify: “Site was included as a random effect (sitej) to account for the uneven sample size. The significance of fixed effects was estimated by Wald`s χ2 test (considering that fixed effects had only two levels, the results also apply to their comparison).”
Point 4
Similarly, it is not at all clear how the comparisons in lines 128-120 of root-plate sizes between soil types was performed. The authors refer to Table 1, but this table does not compare those two variables.
Response 4
See the response above.
Point 5
The manuscript requires a significant revision in terms of written English.
Response 5
The language of the manuscript was revised by professional English editing service.
Point 6
The dependent variable in Equation (1) (line 83) should be “VROOTS” and not just “V”
Response 6
Corrected
Point 7
References should be completed, particularly numbers 4, 14, 16, 20, 24, and 33
Response 7
The mentioned references look different because they are not journal articles, but books, reports, websites, book sections, thus their formatting is different from journal article format. These formats were generated according to the journal guidelines in the programme Mendley.

Reviewer 2 Report
Dear Authors,
As per my view, manuscript is coming under the scope of the journal and having novelties. But at this moment manuscript is not suitable for publication and review process because of a lot of shortcoming such as state of the art is missing in the introduction section, gaps is not mentioned clearly, methodologies used in the study is good, result and discussions are poorly written, implementation/ comperative table is missing, therefore manuscript could be revised carefully.
Author Response
Reviewer 2
Dear Authors,
As per my view, manuscript is coming under the scope of the journal and having novelties. But at this moment manuscript is not suitable for publication and review process because of a lot of shortcoming such as state of the art is missing in the introduction section, gaps is not mentioned clearly, methodologies used in the study is good, result and discussions are poorly written, implementation/ comparative table is missing, therefore manuscript could be revised carefully.
Response
We agree that there were some shortcomings in the manuscript as shown in detail by other reviewers accordingly we have amended them. The main revisions touched introduction and justification of the necessity of the study. Some clarification in methods have been provided.
Unfortunately, the comments of this reviewer appeared too general and contradict the marks in the evaluation chart (e.g., regarding methodology). Also, we were not able to understand the necessity of the comparative table. We, however, now provide more details of factors of fixed effects and by doing so, we alleviate the necessity for Post-Hoc tests, which are useless for effects of two levels (if this is what reviewer had in mind).

Reviewer 3 Report
A brief summary
The manuscript entitled: „Soil type rather than freezing determines the size of soil-root plate of Silver Birch (Betula pendula Roth.) in the Eastern Baltic Region” is very interesting. This manuscript describes relationships between soil-root plate properties and environmental conditions. The manuscript is coherent and well written. It contributes to enhance the knowledge on this topic. Based on this general evaluation and the specific comments, reported below. I recommend a minor revisions of the manuscript and re-write before it will be acceptable for publication. I have few specific comments, which might improve the manuscript.
Methods
Equation [1] and [2] – are these equations constructed by the Authors. Equation [1] used in work Krišāns et al. 2022. Maybe an explanation should be given; Equation [1] according previous work Krišāns et al. 2022.
Equation [2] - how this equation was constructed? Maybe on the basis of previous work? Please, clarify.
Author Response
Reviewer 3
Point 1
Equation [1] and [2] – are these equations constructed by the Authors. Equation [1] used in work Krišāns et al. 2022. Maybe an explanation should be given; Equation [1] according previous work Krišāns et al. 2022.
Response 1
The choice for applying Equation [1] is based on visual observation of shape un overturned soil-root plates. The reference is now given.
Point 2
Equation [2] - how this equation was constructed? Maybe on the basis of previous work? Please, clarify.
Response 2
The construction of Equation [2] was based on regional (Latvian) characteristics of stem shapes of certain forest tree species. In this case, silver birch. The reference is now given.

Round 2
Reviewer 1 Report
1.- In relation to the statistical model:
- the model proposed by the authors is logically wrong whatever is the focus of their research: in the context of their research root volume does not depend on type of failure but type of failure may, in principle, depend on root volume. Therefore, in any model including both variables, root volume should be an explanatory variable and type of failure should be a response variable and not the other way round,
- furthermore, if according to the statement by the authors “trees break rather than uproot under frozen soil conditions” (lines 43-44), then these two variables cannot be used as explanatory variables at the same time as they are not independent,
- the authors state that the results of the logistic model that they show in their response are “trivial and expected”. One such result is that the size of the root plate does not influence the type of tree failure:
- if the authors already expected such lack of relation they should not have used type of failure as a variable in their model,
- but the authors state that “the difference in soil-root plate size for trees suffering uprooting or stem breakage have not been investigated” (lines 44-45), and if this is the case, the results are neither trivial nor expected,
- in any of the cases, type of failure may be removed from the model. But the authors should make their minds up about what their a priori stance was, explain it clearly in the manuscript, and develop their statistical models accordingly,
- the authors frequently discuss the interaction between soil type and soil condition (e.g., lines 114-116; Figure 1). But this interaction has not been included in the model. The authors should do so,
- lines 102-104 added in the new manuscript do not clarify the use of statistical packages. Certainly these packages may complement each other but the purpose of each one should be clearly spelt out
- the sentence added in lines 48-49 does not add anything new to the paper, as artificial uprooting of broken trees had already been reported in lines 78-79.
2.- References 4, 14, 20, 21, 32, 33 do not follow the journal guidelines. References 16 and 24 have been properly corrected.
Author Response
Reviewer 1
Point 1
The model proposed by the authors is logically wrong whatever is the focus of their research: in the context of their research root volume does not depend on type of failure but type of failure may, in principle, depend on root volume. Therefore, in any model including both variables, root volume should be an explanatory variable and type of failure should be a response variable and not the other way round.
Response 1
We disagree, from the statistical point of view there is no difference, which is the response variable, as statistical procedures are the way to support the idea, hence narrative. Accordingly, statistical analysis should serve the focus of research. We agree that the proposed model in this form would be awkward for assessment of factors affecting type of failure. However, we have previously explicitly stated that the model was used to assess the differences for which it is totally fine. Support this cause we now clarify the text.
Lines 93–95, we now clarify and state: “The differences in soil-root plate dimensions between trees experiencing stem breakage or uprooting, under frozen/non-frozen soil conditions, and according soil type were estimated using a linear mixed-effects model:…”
Point 2
Furthermore, if according to the statement by the authors “trees break rather than uproot under frozen soil conditions” (lines 43-44), then these two variables cannot be used as explanatory variables at the same time as they are not independent,
Response 2
The collinearity of those two was checked by the variance of inflation factor; however, there was none. Accordingly, this suggests that they are statistically independent, regarding the dimensions of soil-root plate. Also, the statement does not imply that stem breakage would occur exclusively under frozen soil conditions, implying independence.
Lines 43–44, we now reword: “Although it is a well-known fact that trees tend to break more frequently under frozen soil conditions [8,9]...”
Point 3
The authors state that the results of the logistic model that they show in their response are “trivial and expected”. One such result is that the size of the root plate does not influence the type of tree failure: if the authors already expected such lack of relation they should not have used type of failure as a variable in their model,
Response 3
If the logical model assessing failure type as the response would be used, this would imply much more testing of environmental variables considering effects of different scale (microrelief, tree, stand, landscape, etc.), which was not our aim to begin with. Indeed, such information is desirable in the community; however, previous attempts to answer which trees will be damaged and how have not got reasonable advance. Under such conditions, we realized that the scope (data set) of our study is small, hence such conclusions cannot be drawn. Accordingly, we uprooted trees after they experienced stem breakage and compared the size of soil-root plate with initially uprooted trees.
Point 4
But the authors state that “the difference in soil-root plate size for trees suffering uprooting or stem breakage have not been investigated” (lines 44-45), and if this is the case, the results are neither trivial nor expected,
Response 4
Once again, the idea of this study was to assess the size of soil-root plate for the trees that experienced stem breakage, hence soil-root plate has not been exposed. Accordingly, we artificially uprooted remaining parts of stem.
Lines 44–45, we now state: “…the soil-root plate size for trees suffering stem breakage have not been investigated.”
Point 5
In any of the cases, type of failure may be removed from the model. But the authors should make their minds up about what their a priori stance was, explain it clearly in the manuscript, and develop their statistical models accordingly,
Response 5
We used the model to test the differences according to three fixed effects studied, and returned the results accordingly.
Point 6
The authors frequently discuss the interaction between soil type and soil condition (e.g., lines 114-116; Figure 1). But this interaction has not been included in the model. The authors should do so,
Response 6
Indeed, the assessment of interaction would be meaningful; however, due to limited sample size they were not tested.
Line 101, we now state: “Due to limited sample size, the interactions were not tested.”
Accordingly, we do not discuss them in the text.
Lines 114–117, we now state: “Moreover, under non-frozen soil conditions birch experienced more frequent stem breakage (exceeding uprooting) on peat soil, suggesting stronger soil-root anchorage due to larger and wider soil-root plates.”
Point 7
Lines 102-104 added in the new manuscript do not clarify the use of statistical packages. Certainly, these packages may complement each other but the purpose of each one should be clearly spelt out
Response 7
Lines 103–105, we now clarify: “Data statistical analysis was done in R software (version 4.1.0.) [19], using the packages “MuMIn” (model evaluation) [20], “emmeans” (comparison of levels of significant effects) [21], and “lme4” (model fit) [22].”
Point 8
The sentence added in lines 48-49 does not add anything new to the paper, as artificial uprooting of broken trees had already been reported in lines 78-79.
Response 8
Smaller line numbers usually come before the larger ones.
Point 9
References 4, 14, 20, 21, 32, 33 do not follow the journal guidelines. References 16 and 24 have been properly corrected.
Response 9
Corrected.

Reviewer 2 Report
Dear authors,
thanks for your revision but still I am not happy with the introduction section, need to be further strengthened using recent literature followed by research gaps and objective preparation.
result and discussion again need to strengthen which types of soil is more important to determine the soil root plate of silver birch
please revise it carefully before acceptance
Author Response
Reviewer 2
Point 1
Thanks for your revision but still I am not happy with the introduction section, need to be further strengthened using recent literature followed by research gaps and objective preparation.
Response 1
To authors` knowledge, most recent relevant literature has been cited. Considering that manuscript was intended as brief report, extensive citation was avoided. Nevertheless, the novelty of the study is now highlighted.
Lines 43–45, we now state: “Although it is a well-known fact that trees tend to break more frequently under frozen soil conditions [8,9], to the authors’ knowledge, the soil-root plate size for the trees suffering stem breakage have not been investigated.”
Point 2
Result and discussion again need to strengthen which types of soil is more important to determine the soil root plate of silver birch. Please revise it carefully before acceptance.
Response 2
The idea of this study was to assess the size of soil-root plate for the trees that experienced stem breakage, hence soil-root plate has not been exposed. Accordingly, we artificially uprooted remaining parts of stem. Therefore, the idea was not to estimate “the importance” of soil type on the size of soil-root plate. To our knowledge, importance analysis is a statistical method used for data mining and hence data reduction in case of extensive data sets, which by no means could be attributed to this study.
Nevertheless, lines 143–146, we now add and state: “Such effect was apparently weaker on peat soil due to wider soil-root plate and shallower soil freezing depth (Table 2), implying lower uncertainties, while supporting suitability of birch on unstable soils under increasing wind effects [31].”

Round 3
Reviewer 1 Report
Thank you for your helpful Response 8. I had never thought of it.
Author Response
We apologize about unnecessarily harsh response from us during previous round.